# Study on Electrical and Mechanical Properties of Double-End Supported Elastic Substrate Prepared by Wet Etching Process

**DOI:** 10.3390/mi15070929

**Published:** 2024-07-20

**Authors:** Ding Song, Wenge Wu

**Affiliations:** School of Mechanical Engineering, North University of China, Taiyuan 030051, China; b200206@st.nuc.edu.cn

**Keywords:** wet etching technology, 304 stainless steel substrates, multi-parameter coupling, XPS and nanoindentation

## Abstract

Preparing elastic substrates as a carrier for dual-end supported nickel chromium thin film strain sensors is crucial. Wet etching is a vital microfabrication process widely used in producing microelectronic components for various applications. This article combines lithography and wet etching methods to microprocess the external dimensions and rectangular grooves of 304 stainless steel substrates. The single-factor variable method was used to explore the influence mechanism of FeCl_3_, HCl, HNO_3_, and temperature on the etching rate, etching factor, and etching surface roughness. The optimal etching parameter combination was summarized: an FeCl_3_ concentration of 350 g/L, HCl concentration of 150 mL/L, HNO_3_ concentration of 100 mL/L, and temperature of 40 °C. In addition, by comparing the surface morphology, microstructure, and chemical and mechanical properties of a 304 stainless steel substrate before and after etching treatment, it can be seen that the height difference of the substrate surface before and after etching is between 160 μm and −70 μm, which is basically consistent with the initial design of 0.2 mm. The results of an XPS analysis and Raman spectroscopy analysis both indicate that the surface C content increases after etching, and the corrosion resistance of the surface after etching decreases. The nano-hardness after etching increased by 26.4% compared to before, and the *ζ* value decreased by 7%. The combined XPS and Raman results indicate that the changes in surface mechanical properties of 304 stainless steel substrates after etching are mainly caused by the formation of micro-nanostructures, grain boundary density, and dislocations after wet etching. Compared with the initial rectangular substrate, the strain of the I-shaped substrate after wet etching increased by 3.5–4 times. The results of this study provide the preliminary process parameters for the wet etching of a 304 stainless steel substrate of a strain measuring force sensor and have certain guiding significance for the realization of simple steps and low cost of 304 stainless steel substrate micro-nano-processing.

## 1. Introduction

ASTM304 stainless steel is widely used in strain force sensors for intelligent cutting tools due to its excellent wear resistance, high-temperature resistance, and high compatibility with cutting tools [1,2,3]. As a type of surface microfabrication technology, etching technology is mainly divided into dry etching and wet etching according to different processing methods and processes. Compared with dry etching, wet etching is widely used in the field of micro- and nano-processing due to its simple operation and low cost [4,5]. With the continuous development of microelectromechanical systems (MEMSs), the position of wet etching technology in the microelectronics industry and aerospace field has become increasingly important. The performance of wet etching mainly depends on the concentration of the etching solution, the ratio of the etching solution, the temperature of the etching solution, and the etching time. The differences in these factors can lead to significant differences in the effect after etching [6,7,8,9]. As early as the 1990s, W Eideloth et al. investigated the wet etching process of gold films and calibrated the etching rate and electrical resistivity of the gold film after wet etching [10]; Y. B. Wu et al. proposed an effective solution for selective etching of ultra-thick copper sacrificial layers to achieve three-dimensional suspended metal microstructures, where the thickness of sacrificial copper laminates can reach over 100 μm [11]. Researchers such as Jinyu Zhang combined photolithography and wet chemical etching to explore the variation of surface microstructure and etching depth of carbon steel with etching time and proposed a geometric prediction model for the surface microstructure of carbon steel [12]. Haydar A. H. Al-Ethari and other researchers studied the effects of processing temperature, processing time, and cold working on the metal removal rate and surface smoothness of stainless steel 420 chemically processed alloy samples using an acid mixture (H_2_O + HCl + HNO_3_ + HF + HCOOH) as an etchant [13]. Hyung Min Lee et al. studied the corrosion rate of STS430 stainless steel under the influence of the etchant stirring rate, temperature, and Fe^3+^ ion concentration. The results showed that the corrosion rate followed a pseudo-first-order reaction equation [14]. Fei XIN et al. studied the effects of initial width, etching time, spray pressure, and etchant temperature on the etching rate of stainless steel through experiments and a mathematical analysis [15,16]. Jae-Hun Kim et al. proposed a two-step chemical etching method of sequentially immersing a stainless steel substrate in hydrochloric acid and ferric chloride solutions to prepare micro-nanostructures on the surface of the stainless steel substrate and form a superhydrophobic surface [17]. Researchers such as Wei investigated the effects of an etching solution and etching time on the etching accuracy of stainless steel [18]. They analyzed the passivation phenomenon that occurs during the etching process. These high-quality research results all indicate that wet etching technology has advantages such as high selectivity, pure chemistry, and low cost in metal microprocessing and multi-layer metal micro-nanostructure preparation.

For applying 304 stainless steel as a substrate for strain gauges, previous research on wet etching stainless steel has focused on understanding the etching rate as a single performance. Few articles have conducted in-depth and systematic research on the characterization of etching quality, etching rate, and various properties after etching. This article is based on the control variable method. It explores in detail the influence trend and mechanism of the coupling effect of multiple process parameters such as FeCl_3_ concentration, HCl concentration, HNO_3_ concentration, and etching temperature on the etching rate, etching factor, and roughness of a 304 stainless steel substrate; it evaluates the surface morphology, microstructure, chemical properties, mechanical properties, and electrical properties of a 304 stainless steel substrate before and after etching. The obtained results provide certain research value and guiding significance for the performance research of wet etching of austenitic stainless steel and the manufacturing and development of related devices.

## 2. Materials and Methods

### 2.1. Test Samples, Reagents, and Testing Instruments

Table 1 and Table 2 show specific information on the test samples, etching reagents, and analytical instruments used in this experiment. This article’s main testing content includes the quality, etching factor, surface roughness, macroscopic and microscopic morphology, XPS electron spectroscopy, Raman spectroscopy, and nano-hardness of a 304 stainless steel substrate before and after wet etching.

### 2.2. Test Method

#### 2.2.1. Etching Rate, Etching Factor, and Etching Surface Roughness

The etching rate, etching factor, and etching surface roughness are essential indicators for evaluating the quality of wet etching. The quality method determines the etching rate based on the difference between the weight of 304 stainless steel before wet etching and the weight of 304 stainless steel after wet etching per unit time. Due to its simple and convenient operation and high reliability, the quality method is usually the preferred method for researchers to evaluate the etching rate. Its calculation formula is
*v* = (*m*_0_ − *m*_1_)/*ρ S t*(1)

The *m*_0_ and *m*_1_ in Formula (1) represent the mass of the stainless steel substrate before and after etching, respectively; *ρ* is the density of the stainless steel substrate; *S* is the area of the etched area; and *t* is the etching time. 

Wet etching is usually isotropic, resulting in a side etching phenomenon. The etching factor is the leading indicator for evaluating the side etching phenomenon [19,20]. As shown in Figure 1, the etching factor is defined as follows:
Z = *D*/[(*W*_1_ − *W*_2_)/2] (2)

In Formula (2), *W*_1_ is the initial width of the upper surface of the stainless steel substrate, *W*_2_ is the final width after the etching of the stainless steel substrate, and *D* is the etching depth.

The surface roughness of 304 stainless steel before and after etching was tested using the optical contour measurement method. A Contour GTK optical contour analyzer was used to measure 304 stainless steel samples under different etching conditions. Five 10 mm long contours were tested for each sample, with a spacing of 1 mm between them, and the average value was calculated to calculate the surface roughness of 304 stainless steel under different etching conditions.

#### 2.2.2. Macroscopic and Microscopic Morphology Diagrams

We observed the macroscopic morphology of the 304 stainless steel surface before and after etching using a 2D planar mode of a VHX-600 ultra-depth optical microscope produced by Keyence Corporation in Osaka, Japan; we observed the microstructure of the 304 stainless steel surface before and after etching using an OLS5000-SFA confocal microscope produced by Olympus Corporation in Tokyo, Japan; in the secondary electron mode, the surface microstructure of 304 stainless steel before and after etching was observed using a Tescan Mira4 scanning electron microscope produced by Hitachi in Tokyo, Japan.

#### 2.2.3. XPS Electron Spectroscopy

The 250Xi X-ray photoelectron spectrometer produced by Thermo Fisher Scientific Inc. in Waltham, MA, USA was used, with a test beam spot size of 500 microns, a full spectrum passing energy of 100 eV, a step size of 1.0 eV, a narrow spectrum passing energy of 30 eV, a step size of 0.1 eV, and a monochromatic Al target Kα line as the X-ray excitation source. An Ar ion beam was used to analyze the elements at 0 nm and 5 nm on the surface of 304 stainless steel before and after etching, with a sputtering rate of 1 nm/min.

#### 2.2.4. Raman Spectra

The LabRAM Odyssey high-speed, high-resolution confocal Raman spectrometer produced by HORIBA Corporation in Kyoto, Japan is used, with a spectral resolution of 0.35 cm^−1^ to 0.65 cm^−1^, a wavelength of 633 nm (Ar + laser) as the excitation light source, and a Raman shift range of 1000–2000 cm^−1^.

#### 2.2.5. Nano-Hardness

The NanoTest Vantage, a comprehensive testing system for mechanical properties of nanomaterials from MML in the UK, is used. The probe is a triangular diamond stone with an angle of 120° between its three faces. The blunt radius of the indenter is less than 200 nm, and Young’s modulus and Poisson’s ratio of the indenter are 1141 GPa and 0.07, respectively. The loading and unloading rates are both 5 mN/s. Before testing, the diamond indenter is calibrated with a quartz standard sample. To ensure the accuracy and repeatability of the testing, we tested each sampling point 6 times.

### 2.3. Specific Processing Flow

The processing flow of wet etching 304 stainless steel substrate is shown in Figure 2.

(a)The I-shaped 304 stainless steel substrate can achieve an 8 K mirror surface by using rough, semi-precision, and precision polishing. The processed stainless steel substrate is sequentially placed in acetone and anhydrous ethanol for ultrasonic cleaning to remove excess impurities on the surface and improve the accuracy of the experiment.(b)Use the homogenizer to spin the photoresist on the substrate of 304 stainless steel at the speed of 2000 r/min.(c)Bake on a Labtech-EH20B hot plate (Laibotaiko Co., Ltd., Beijing, China) at 100 °C for 100 s, attach the designed film mask to the surface of 304 stainless steel coated with the photoresist, and then expose it using an EVG610 lithography machine and develop it in the developing solution for 100 s, followed by rinsing with deionized water for 35 s.(d)Bake on a Labtech-EH20B hot plate at 120 °C for 600 s, and finally perform a microscopic examination using a laser confocal microscope to observe if its microstructure meets the requirements.(e)Place the 304 stainless steel substrates wrapped with the photoresist in the configured etching solution using a clamping tool for etching. Control the reaction temperature through a constant temperature heating table and continuously adjust the ratio of the etching solution, etching temperature, and etching time to obtain the required process parameters.(f)Peel off the photoresist wrapped on the surface of 304 stainless steel after wet etching using acetone and anhydrous ethanol solutions, and perform a microscopic examination using a laser confocal microscope to observe whether the overall shape of the wet-etched stainless steel substrate conforms to the expected design and whether the etching depth and surface flatness are within a reasonable range.

## 3. Results and Analysis

### 3.1. Etching Results and Mechanisms

Many experiments were conducted using the single-variable method, and the experimental data were integrated. Finally, a bar chart of the etching rate, etching factor, surface roughness, and variables such as FeCl_3_, HCl, HNO_3_, and temperature, as shown in Figure 3, Figure 4, Figure 5 and Figure 6, was plotted. 

The concentrations of FeCl_3_ in the prepared etching solution were 50, 150, 250, 350, and 450 g/L, respectively. The concentration of HCl was 100 mL/L, and the concentration of HNO_3_ was 100 mL/L. The etching temperature was 40 °C, and the results are shown in Figure 3. Figure 3 shows that during the wet etching process, with the continuous increase in FeCl_3_ concentration, the etching rate first rapidly increases and then slowly decreases. This is because at the beginning of the reaction, with the increase in FeCl_3_ concentration in the etching solution, the Fe^+3^ per unit area of the stainless steel surface increases, and there is more Fe^+3^ per unit area reacting with the metal M (Mn ≤ 2%, Ni = 8–11%, Cr = 18–20%, Fe = 68–70%) in the stainless steel, which promotes the increase in the entire reaction rate and the etching rate rapidly increases. Because at the beginning of the reaction, with the increase of FeCl_3_ concentration in the etching solution, the Fe^+3^ per unit area of the stainless-steel surface increases, and more Fe^+3^ per unit area reacts with the metal M in stainless steel (Mn ≤ 2%, Ni = 8–11%, Cr = 18–20%, Fe = 68–70%). Accelerate the whole reaction, the etching rate increases rapidly. However, as the Fe^+3^ concentration further increases, excessive Fe^+3^ in the solution will affect its migration, which is not conducive to the occurrence of the entire reaction. The etching rate will decrease to a certain extent [21]. With the continuous increase in FeCl_3_ concentration, the etching factor first slowly increases and then gradually decreases. The etching factor mainly depends on the ratio of the vertical etching rate to horizontal etching rate. The larger the value, the smaller the sidewall etching and the better the etching effect. During the initial reaction period, as the FeCl_3_ concentration in the etching solution increases, the vertical etching rate is faster than the horizontal etching rate, and the etching factor gradually increases. However, as the FeCl_3_ concentration in the etching solution further increases, the vertical etching rate slows down compared to the horizontal etching rate, and the etching factor decreases reversely. The roughness value first increases rapidly and then slowly decreases with the increase in FeCl_3_ concentration. When the FeCl_3_ concentration is low, some areas of the stainless steel surface have lower potential barriers and are easy to etch. The etched product particles are small and have lower roughness. However, as the FeCl_3_ concentration increases, the etched product on the stainless steel surface is coarse and disorderly. This results in the roughness reaching the maximum value of 8.22 μm at an FeCl_3_ concentration of 250 g/L. As the concentration of FeCl_3_ further increases, the products on the surface of stainless steel after etching gradually increase and become dense, resulting in a decrease in overall roughness and an improvement in surface quality. 

The concentrations of HCl in the prepared etching solution were 0, 50, 100, 150, and 200 mL/L, respectively. The concentrations of FeCl_3_ and HNO_3_ were all 350 g/L and 100 mL/L, respectively, and the etching temperature was 40 °C; the results are shown in Figure 4. Figure 4 shows that during the wet etching process, with the continuous increase in HCl concentration, the etching rate shows a trend of first increasing and then decreasing. This is because as the concentration of HCl continues to increase, the pH value in the solution decreases while the H^+^ in the solution increases. On the one hand, the H^+^ in the solution also has an etching effect on iron in stainless steel. On the other hand, as the pH value of the solution decreases, it can effectively inhibit the hydrolysis of Fe^3+^ into Fe (OH)_3_, increase the etching ability of the solution, and improve its etching rate. FeCl_3_ has up to six vacant orbitals. In FeCl_3_·6H_2_O, six water molecules are distributed in octahedra and coordinate with Fe^3+^ ions. When excessive HCl is added to an FeCl_3_ solution, the high concentration of Cl^−^ in the solution forms complexes with Fe^3+^ in the solution, such as HFe(H_2_O)Cl_4_, H_2_Fe(H_2_O)Cl_5_, and H_3_FeCl_6_, indirectly providing a certain protective effect on the stainless steel substrate, leading to a decrease in the etching rate. With the continuous increase in HCl concentration, the etching factor increases and shows a basic positive correlation; this is because, during the etching process, the vertical etching rate of the stainless steel substrate increases significantly more than the horizontal etching rate, leading to a continuous increase in the etching factor with the increase in HCl concentration. The roughness decreases continuously with the increase in HCl concentration and shows a basic negative correlation. This is because, with the continuous increase in HCl concentration, the concentration of Cl^−^ in the etching solution increases. Cl^−^ can form complexes FeCl_4_^−^ with Fe^3+^ metal ions. At the same time, the continuously generated Fe^2+^, Cr^3+^, and Ni^2+^ ions in the solution are challenging to diffuse on the surface of stainless steel, causing the trend of atoms on the surface of stainless steel becoming corresponding ions to slow down. This phenomenon is particularly evident in areas with fast etching rates, such as areas with severe pitting, and is not obvious in non-pitting areas. Therefore, the overall performance is that the etching surface becomes smooth and flat, and the roughness decreases continuously. 

The concentrations of HNO_3_ in the prepared etching solution were 0, 50, 100, 150, and 200 mL/L, respectively. The concentrations of FeCl_3_ and HCl were all 350 g/L and 100 mL/L. The etching temperature was 40 °C, and the results are shown in Figure 5. Figure 5 shows that during the wet etching process, with the continuous increase in HNO_3_ concentration, the etching rate continues to increase, especially when the HNO_3_ concentration is higher than 100 mL/L, and the increase in the etching rate is particularly significant. This is because HNO_3_, as a highly oxidizing acid, can significantly accelerate the reaction between FeCl_3_ and stainless steel when acting as an oxidant in an FeCl_3_ solution. Therefore, as the concentration of HNO_3_ increases, the etching rate continues to increase. However, high concentrations of HNO_3_ are prone to dissociation, which has a certain impact on the stability of the etching solution. At the same time, the etching rate is too fast, which is also not conducive to the controllability of the etching process. As the concentration of HNO_3_ continues to increase, the etching factor continues to increase and shows a positive correlation, similar to the effect of HCl on the etching factor. The roughness increases with the increase in HNO_3_ concentration, and etching occurs under certain conditions when the complex oxides on the surface of stainless steel are destroyed, and especially the inclusion points of defects such as sulfides, oxides, carbides, etc., are more prone to occur. As HNO_3_ is a strong oxidizing acid, with the addition of HNO_3_, the etching rate on the surface of stainless steel continues to increase. The etching phenomenon in the defective areas that have been improved above becomes more obvious, resulting in uneven etching in all directions. The macroscopic manifestation is that the roughness increases continuously with increased HNO_3_ concentration. Therefore, while ensuring the etching rate, low concentrations of HNO_3_ should be selected as much as possible.

The temperature in the prepared etching solution is 30, 35, 40, 45, and 50 °C, respectively. The concentration of FeCl_3_ is 350 g/L, the concentration of HCl is 100 mL/L, and the concentration of HNO_3_ is 100 mL/L. The results are shown in Figure 6. Figure 6 shows that during the wet etching process, as the etching temperature continues to increase, the etching rate shows a trend of initially increasing, which is positively correlated. On the one hand, as the temperature of the etching solution increases, the activation energies of various ions such as Fe_2_^+^, Cr_3_^+^, Ni_2_^+^, Cl^−^, NO_3_^−^, and H^+^ in the solution increase, and the reaction rate correspondingly increases. On the other hand, as the temperature of the etching solution increases, the molecular thermal motion increases, accelerating the fluidity of the etching solution, which is conducive to improving the diffusion rate of reactants and reaction products. Therefore, the etching rate of the etching solution continuously increases with the temperature of the etching solution. Although raising the temperature can significantly improve the etching rate, due to a certain concentration of HNO_3_ in the etching solution, too high a temperature will accelerate the reaction of HNO_3_ with stainless steel to generate a large amount of yellow irritating gas NO, etc. The environment causes certain damage, and excessive temperature can exacerbate the detachment of the photoresist on the unexcavated surface of stainless steel, which is not conducive to the continuation of subsequent etching reactions. Therefore, considering all factors while ensuring the etching rate, the temperature should not be too high. As the etching temperature continues to rise, the etching factor first increases rapidly and then slowly decreases. This may be due to the high etching temperature, which causes the photoresist on the surface of stainless steel to dissolve and peel off, decreasing the etching factor [22]. Therefore, the reaction temperature should not be too high; as the etching temperature continues to rise, the roughness first increases rapidly and then tends to stabilize. Because the etching temperature continues to rise, the etching rate continues to rise, leading to a continuous increase in etching products. The micro-pits and pitting phenomena on the surface of the stainless steel substrate continue to intensify, resulting in a continuous increase in the roughness value of the surface [23]. Therefore, it is crucial to control the etching temperature reasonably.

Therefore, under the experimental conditions of this article, taking into account the influence trend and mechanism of various experimental factors (FeCl_3_ concentration, HCl concentration, HNO_3_ concentration, and etching temperature) on the experimental results (etching rate, etching factor, and roughness of stainless steel), the process conditions for wet etching were selected as follows: FeCl_3_ concentration of 350 g/L, HCl concentration of 150 mL/L, HNO_3_ concentration of 100 mL/L, and etching temperature of 40 °C. 

### 3.2. Microstructural Analysis

#### 3.2.1. Surface Morphology Analysis

Local wet etching was performed on the 304 stainless steel substrate using optimized process conditions. Figure 7 shows the surface morphology and etching depth of the wet-etched 304 stainless steel substrate. Figure 7a shows the macroscopic surface of the locally etched 304 stainless steel substrates taken by the VHX-600 ultra-depth field optical microscope, and Figure 7b shows the three-dimensional scanning image of the black small square area in Figure 7a taken by the OLS5000-SFA confocal microscope. The area of the black small square area is close to a square with a side length of 2561 μm; Figure 7c is a plan view of the three-dimensional scanning of the black small square area, and Figure 7d is a schematic diagram of the height difference of the small black square area. Figure 7a shows that before wet etching, the surface of the stainless steel substrate has good uniformity, high flatness, and fewer surface scratches. After wet etching, the grain size of the surface of the 304 stainless steel substrate is relatively consistent, and there are no obvious protrusions or depressions on the surface. From Figure 7b–d, it can be seen that after wet etching, the height variation range of the nonetched and etched surfaces of the 304 stainless steel substrate is between 160 μm and −70 μm, excluding the influence of some local protrusions and depressions. It is basically consistent with the initial groove depth of 0.2 mm, which meets the design requirements. In addition, through the fluctuation of the curves of the unetched and etched surfaces in Figure 7d, we can observe that the curve at the connection between the unetched and etched surfaces is not vertical but has a gentle slope trend. There may be two possible reasons for this. The first point may be that the wet etching technique used in this article for etching stainless steel is isotropic, and the second point may be that the etching at the connection between the unetched and etched surfaces is incomplete, with some remaining unetched.

Figure 8 is a confocal microscope image of the 304 stainless steel substrate surface before and after wet etching treatment. Figure 8a shows that the surface of the 304 stainless steel substrates without wet etching treatment is basically undamaged, and the microstructure remains unchanged. From Figure 8b, it can be seen that there is much minor pitting on the surface of the wet etching-treated stainless steel, forming a lattice sand surface, and the grains and grain boundaries begin to be exposed.

Scanning electron microscopy was performed on the sample under the secondary electron mode further to analyze the microstructure characteristics of the sample surface. Figure 9 shows the scanning electron microscopy image of the 304 stainless steel substrate surfaces before and after wet etching treatment, with the inset in the upper right corner displaying a low magnification SEM image (1 μm). The lower left illustration shows a high-magnification SEM image (500 nm). Figure 9a shows that after higher magnification, a few micro-pits appeared on the surface of the original 304 stainless steel substrate, which may be due to the surface not being polished and containing some defects. However, observing the confocal microscope and scanning electron microscope images shows that the microstructure of the surface of the 304 stainless steel substrates without wet etching treatment still remains unchanged. Figure 9b shows that the surface of the 304 stainless steel substrates treated with wet etching exhibits a grain-like microstructure, with significant corrosion occurring at both the grain and grain boundaries, increasing roughness and porosity. This is because high-energy sites, such as dislocations and grain boundaries, always exist in the crystalline metal, and due to their high energy, they are more susceptible to etching agent attacks and will dissolve first [24,25]. Therefore, this selective etching behavior will generate certain roughness and voids on the surface of the etched stainless steel. Throughout the entire wet etching process, this etching advantage has always existed. The depth and diameter of nanostructures in pitting or micro-pits increase continuously as the reaction continues. Therefore, after etching with an FeCl_3_ solution, micro-nanostructures with micron-scale roughness will be formed. 

#### 3.2.2. Chemical Characteristic Analysis

To compare and analyze the elemental states on the surface of the 304 stainless steel substrate before and after wet etching, the total spectrum at 0 nm and 5 nm on the surface of 304 stainless steel before and after etching, as well as the spectra of elements such as C, O, N, Fe, Ni, and Cr, was analyzed. The XPS spectrum was corrected for charge using the standard peak of C1s (284.8 eV).

Figure 10 shows the total spectrum and respective spectrograms of different elements. The total spectrum in Figure 10 shows that the surface of 304 stainless steel substrates contains elements such as C, O, N, Fe, Ni, and Cr. A comparative analysis shows that the C and O content at 0 nm on the surface of the 304 stainless steel substrate before and after wet etching is significantly higher than that at 5 nm. The possible reason is that the surface of the 304 stainless steel substrate is exposed to air for a long time for oxidation and pollution. After removing the polluted elements, such as C, O, N, Fe, Ni, and Cr, at 5 nm on the surface of the 304 stainless steel substrate, before and after wet etching, there is a return to normal, and especially the content of Fe, Ni, and Cr elements has significantly increased. After the contaminated elements on the surface of 304 stainless steel substrate are removed, for example, after wet etching, the elements such as C, O, N, Fe, Ni, and Cr at 5nm of the surface of 304 stainless steel substrate return to normal, especially the content of Fe, Ni, and Cr has been significantly improved.

From the C1s spectrum in Figure 10, it can be seen that there is a strong C elemental peak and a weak O-C=O at 0 nm and 5 nm on the surface of the 304 stainless steel substrate before and after wet etching, corresponding to binding energies of 284.8 eV and 288.5 eV, respectively. However, compared to the 0 nm on the surface of the 304 stainless steel substrates without wet etching, there is a strong C-O-C peak at 0 nm on the surface of the wet etching-treated stainless steel substrate, with a binding energy of 286 eV, and the C-O peak on the surface of the wet etching-treated 304 stainless steel substrates has increased.

From the O1s spectrogram in Figure 10, it can be seen that a strong metal carbonate peak (532 eV) appears at 0 nm on both the unexcavated and etched surfaces, but an additional metal oxide peak (530 eV) appears at 0 nm on the unexcavated and etched surfaces. Metal carbonate peaks (531.5~531.6 eV) and metal oxide peaks (530 eV) appeared at both the unetched and etched surfaces at 5 nm. However, the metal carbonate peaks (531.5~531.6 eV) at 5 nm on the etched surface increased compared to the unetched surface, which is consistent with the increase in C-O peaks on the 304 stainless steel substrate surfaces after wet etching treatment in the C1s spectrogram.

From the N1s spectrogram in Figure 10, it can be seen that two peaks, N-C (399.7 eV) and metal nitrates (397.2 eV), appear at 0 nm and 5 nm on both the unexcavated and etched surfaces. Based on the main components in 304 stainless steel, it is inferred that the peak at 397.2 eV is Cr2N, and regardless of etching or not, the Cr2N peak at 5 nm on the surface is more potent than that at 0 nm on the surface.

From the Fe2p spectrogram in Figure 10, it can be seen that two Fe2p3/2 peaks (710.7 eV and 706.7 eV) appear at 0 nm on the surface without etching. The characteristic peak of 710.7 eV belongs to Fe_2_O_3_ in the Fe-O bond, and the characteristic peak of 706.7 eV belongs to iron. Two Fe2p3/2 peaks (711.3 eV and 709.6 eV) were observed at 0 nm on the same etched surface. The characteristic peaks of 711.3 eV and 709.6 eV belong to Fe_2_O_3_ and FeO bonds in the Fe-O bond. In addition, a strong Fe elemental peak (706.8 eV) was found at 5 nm on both the unexcavated and etched surfaces. Combined with the characteristic peak of metal nitrides (397.2 eV) in the N1s spectrogram, it can be seen that at 5 nm, the unexcavated surface has an additional FeN peak compared to the etched surface. However, iron nitride compounds target stainless steel. The improvement of corrosion resistance has little effect. Because from the Fe2p spectrogram, it can be seen that the characteristic peaks at 0 nm on the unexcavated and etched surfaces are mainly Fe_2_O_3_ and FeO, In comparison, the characteristic peaks at 5 nm on the unexcavated and etched surfaces are mainly Fe elemental. It can be seen from the Fe2p spectrogram that the characteristic peaks at 0nm on the unexcavated and etched surfaces are mainly Fe_2_O_3_ and FeO. In comparison, the characteristic peaks at 5 nm on the unexcavated and etched surfaces are mainly Fe elemental.

From the Ni2p spectrogram in Figure 10, it can be seen that Ni2p3/2 (852.6~852.8 eV) peaks are present at both 0 nm and 5 nm on both the unexposed and etched surfaces, corresponding to the Ni peak. However, compared to the unexposed and etched surfaces at 0 nm, there is an unusually weak NiO (853.6~853.9 eV) peak at 5 nm on the unexposed and etched surfaces, indicating that the vast majority of Ni exists in the elemental form before and after etching.

From the Cr2p spectrogram in Figure 10, it can be seen that there is a strong Cr2p3/2 (576.6~577 eV) peak and a weak Cr2p3/2 (574~574.1 eV) peak at 0 nm on the surface that has not been etched or etched. The characteristic peaks of 576.6–577 eV belong to Cr_2_O_3_ in the Cr-O bond, and the characteristic peaks of 574–574.1 eV belong to Cr elemental. On the contrary, there is a weak Cr2p3/2 (576.2~576.4 eV) peak and a strong Cr2p3/2 (574~574.2 eV) peak at 5 nm on the surface that has not been etched or etched. Similarly, 576.2~576.4 eV peaks belong to Cr_2_O_3_ in the Cr-O bond, and 5774~574.2 eV characteristic peaks belong to Cr elemental. Based on the O1s spectrogram, it can be seen that Cr mainly exists in the form of Cr oxides at 0 nm on the surface. Combined with the Fe2p spectrogram, it can be seen that during the wet etching process, Fe_2_O_3_ and FeO at 0 nm on the surface quickly detach, and then a dense layer of Cr_2_O_3_ is formed on the surface. Combined with the N1s spectrogram, it can be concluded that the gold nitride has an FeN peak and should be a mixture of FeN and CrN.

The Raman spectroscopic analysis of the 304 stainless steel substrate surface in the etched and nonetched regions is shown in Figure 11. No Raman peaks were observed in the nonetched region, while solid peaks were observed in the etched region at 1288 cm^−1^ and 1609 cm^−1^, respectively. These two peaks are typical Raman peaks of the D and G peaks of carbon elements, with 1288 cm^−1^ mainly induced by the stretching vibration of C-C bonds and C=C double bonds, representing lattice defects; 1609 cm^−1^ is mainly caused by the stretching vibration of sp2 hybridized carbon atoms, but the positions of the D and G peaks in the etching region of this article have shifted (D peak shifted towards low wavenumber, G peak shifted towards high wavenumber). This is due to the shift of Raman peaks being determined by the chemical bonds and molecular structure in the sample [26]. Therefore, a large number of sp2 hybridized carbon atoms and a large number of C-C bonds were generated in the etching region. The peak intensities of the D and G peaks at 1531 cm^−1^ and 1728 cm^−1^ are relatively low, and the XPS analysis suggests that the 1531 cm^−1^ peak is caused by the stretching vibration between M and C/N (metal and C and N). Based on the analysis results of C1s and N1s in XPS, 1728 cm^−1^ is mainly induced by the stretching vibration of C-N bonds. The results of the Raman spectroscopy analysis and XPS analysis confirm each other and indicate that the increase in surface carbon content after etching leads to decreased corrosion resistance of the etched area.

### 3.3. Mechanical Characteristic Analysis

#### 3.3.1. Mechanical Property Analysis 

Figure 12 shows the indentation depth/applied load curve of nanoindentation at a maximum load of 100 mN. According to the loading/indentation depth curve in Figure 12, the unloading curve and the area enclosed by the maximum depth represent the recoverable deformation energy, while the loading curve and the area enclosed by the maximum depth represent the total deformation energy. The ratio *ζ* between them can generally be regarded as a measure of the material’s elastic properties [27]. Record the mechanical properties of the materials tested and calculated by the nanoindentation test in Table 3. From Table 3, it can be seen that the elastic modulus of the wet-etched area of the 304 stainless steel substrate increased by about 22.8% compared to the nonetched area. At the same time, the nano-hardness of the wet-etched area of the 304 stainless steel substrate increased by about 26.4% compared to the nonetched area, and the *ζ* value decreased by about 7%. The increase in the elastic modulus and nano-hardness and decrease in the *ζ* value come from the nanostructures and high-density dislocations generated by wet etching. The increase in grain boundary density and dislocations slows down the movement of dislocations, leading to an increase in the elastic modulus and nano-hardness. Due to the shortening of the elastic deformation range, the decrease in yield strength also reduces the *ζ* value. It can be seen that wet etching treatment has changed the mechanical properties of the surface of 304 stainless steel. Combined with the previous XPS and Raman spectroscopy analysis results, it can be seen that since there is no noticeable composition change, the change in surface mechanical properties after wet etching is mainly caused by the micro- and nanostructures, grain boundary density, and dislocation formed after wet etching. 

#### 3.3.2. Electrical Characteristic Analysis

The I-shaped substrate after wet etching and the initial untreated rectangular substrate are taken as samples through theoretical calculation, simulation analysis, and experimental verification to qualitatively analyze the improvement of the strain performance of the I-shaped substrate after wet etching compared with the untreated rectangular substrate. 

Due to the allowable bending strength of 304 stainless steel being 200 MPa, the maximum pressure that the free end of the rectangular base selected in this article can withstand is 5.6 N, and the maximum pressure that the free end of the I-shaped base prepared can withstand is 2 N. Therefore, a load range of 0.1 N to 1 N is selected for this article. 

During the experiment, strain gauges were pasted at a distance of 15 mm from the applied load. Figure 13a shows the schematic diagram of the base bending test, and Figure 13b shows the testing device for the base bending test. As shown in Figure 13a, fix the I-shaped substrate and rectangular substrate with strain gauges attached to the clamp and apply a load of 0.1 N to 1 N on the free end of the substrate using a tensile and compressive testing machine. Attach a resistance strain gauge on the surface of the substrate to measure the true strain at a distance of 15 mm from the applied load on the substrate, and form a 1/4 bridge circuit with three external resistance strain gauges of the same size. Each resistance strain gauge has a resistance value of 350 Ω and has a temperature self-compensation function. Figure 13b shows that the bending test device consists of ① an elastic substrate with a strain gauge, ② a tensile and compressive testing machine, ③ a DC power supply, and ④ a KEITHLEY2450 digital source meter.

The specific experimental process is as follows:(1)Place the I-shaped substrate and rectangular substrate in acetone and anhydrous ethanol sequentially for ultrasonic cleaning and blow dry with a nitrogen gun to remove any remaining impurities on the surface;(2)Select four 350 Ω resistance strain gauges, paste one of them onto the substrate surface using a modified acrylic adhesive, and connect the remaining three resistance strain gauges to form a 1/4 Wheatstone bridge as shown in Figure 13a;(3)Use a multimeter to check for short circuits or open circuits between the Wheatstone bridge and the resistance strain gauge and substrate;(4)The fixed clamping table, respectively, fixes one end of the I-shaped base and the rectangular base, and the load from 0.1 N to 1 N is applied to the free end of the I-shaped base and the rectangular base, respectively, by the tension–compression testing machine. The KEITHLEY2450 digital source meter and the DC power supply are connected to the Wheatstone bridge’s output and input, and the output voltage of the Wheatstone bridge is measured and recorded.

In order to compare the errors between experimental results, theoretical calculation results, and simulation analysis results, the theoretical strains of the optimized I-shaped substrate and the unoptimized rectangular substrate were calculated using (a) in Formula (3) under loads ranging from 0.1 N to 1 N. Finally, the theoretical calculation results were recorded in Table 4. The strain of the I-shaped base and the unoptimized rectangular base was analyzed using ANSYS finite element simulation software. The materials of the I-shaped base and the unoptimized rectangular base were both 304 stainless steel. Therefore, the elastic modulus was set to 194 GPa, Poisson’s ratio to 0.29, and density to 7.93 g/cm^3^. One section of the I-shaped base and the unoptimized rectangular base was fixed, and a concentrated load of 0.1 N to 1 N was applied to the other end. The high-order ten-node tetrahedral structural element SOLID187, supported by ANSYS software, was used to mesh the three-dimensional solid models of the I-shaped base and the unoptimized rectangular base structure. Finally, the finite element analysis results were recorded in Table 4. Due to the strain sensitivity coefficient k of the resistance strain gauge used being 2.11, by substituting the output voltage and input voltage measured in the experiment to (b) in Formula (3), the experimental strain of the wet etching-treated I-shaped substrate and the untreated rectangular substrate can be calculated. The theoretical, simulation, and experimental values are recorded in Table 4, respectively.
*ε* = *F l Z*/*E*_b_ ∫ *Z*^2^ *d A* (a)     *U*_0_ = 0.5 *k F l Z U_i_*/*E*_b_ ∫ *Z*^2^ *d A* (b) (3)
where *ε* is the strain of the substrate subjected to concentrated force *F*, *E*_b_ is the elastic modulus of the substrate, *l* is the distance from the stress point of the substrate to the fixed end, *Z* is the vertical distance from the surface of the substrate to the *Y* plane of the central layer of the cross-section, *k* is the strain sensitivity coefficient of the resistance strain gauge, *U*_i_ is the input voltage, and *U*_0_ is the output voltage. 

From Table 4, we can see that the theoretically calculated values of the rectangular and I-shaped substrate are greater than the experimental values. This may be due to errors caused by fixing the 304 stainless steel substrate and pasting resistance strain gauges. According to the ratio of the theoretically calculated values, simulation analysis values, and experimental values of the rectangular substrate and I-shaped substrate, it can be seen that the strain of the I-shaped substrate after wet etching treatment is between 3.5 and 4 times that of the rectangular substrate, and the trend of the three ratios is the same. The results of the theoretical calculation and simulation analysis can verify the correctness of the experimental results. Therefore, the elastic substrate treated by wet etching can significantly increase strain, providing a solution for collecting small strains.

## 4. Conclusions

This article is based on the control variable method to explore the influence trend and mechanism of wet etching process parameters on the 304 stainless steel substrate of dual-end supported nickel chromium thin film strain sensors. The electrical and mechanical properties of the etched 304 stainless steel substrate were tested, and the following main conclusions were obtained:(1)The influence of multiple process parameters such as FeCl_3_ concentration, HCl concentration, HNO_3_ concentration, and etching temperature on the etching rate, etching factor, and substrate roughness was analyzed. The reasons for this trend were explained. The optimal wet etching process conditions were selected: an FeCl_3_ concentration of 350 g/L, HCl concentration of 150 mL/L, HNO_3_ concentration of 100 mL/L, and etching temperature of 40 °C.(2)The surface of the substrate before and after etching was macroscopically and microscopically examined using a VHX-600 ultra-deep-field optical microscope, an OLS5000-SFA confocal microscope, and a Tescan Mira4 scanning electron microscope. The height difference before and after etching was between 160 μm and −70 μm, excluding the influence of local protrusions and depressions, which was basically consistent with the initial design of 0.2 mm. In addition, the FeCl_3_ solution will form a micro-nanostructure with micron-scale roughness.(3)The results of the Raman spectroscopy analysis and XPS analysis mutually confirm that the surface C content increases after etching, and the corrosion resistance of the etched surface decreases.(4)According to the results of the nanoindentation test, it can be concluded that the elastic modulus of the substrate before and after etching differs by 22.8%. The nano-hardness after etching is increased by 26.4%, and the *ζ* value is reduced by 7%. Moreover, the combined XPS and Raman results indicate that the changes in the surface mechanical properties of the substrate after etching are mainly caused by the micro-nanostructures, grain boundary density, and dislocations formed after wet etching.(5)Through a comprehensive analysis of theoretical calculations, a simulation analysis, and bending test results, it can be concluded that the strain of the I-shaped substrate after wet etching treatment is 3.5 to 4 times higher than that of the untreated rectangular substrate.

## Figures and Tables

**Figure 1 micromachines-15-00929-f001:**
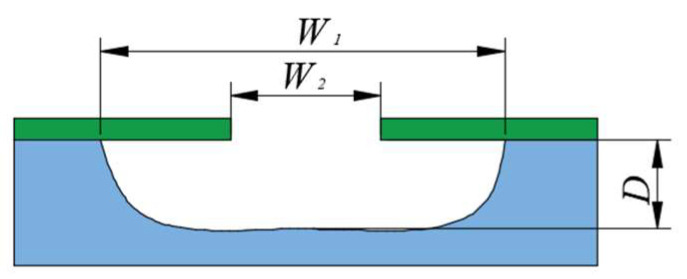
Etching schematic diagram of stainless steel substrate.

**Figure 2 micromachines-15-00929-f002:**
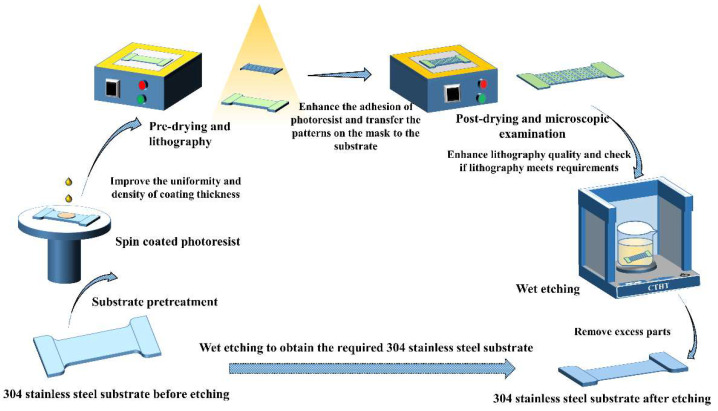
Wet etching process flow of 304 stainless steel substrate.

**Figure 3 micromachines-15-00929-f003:**
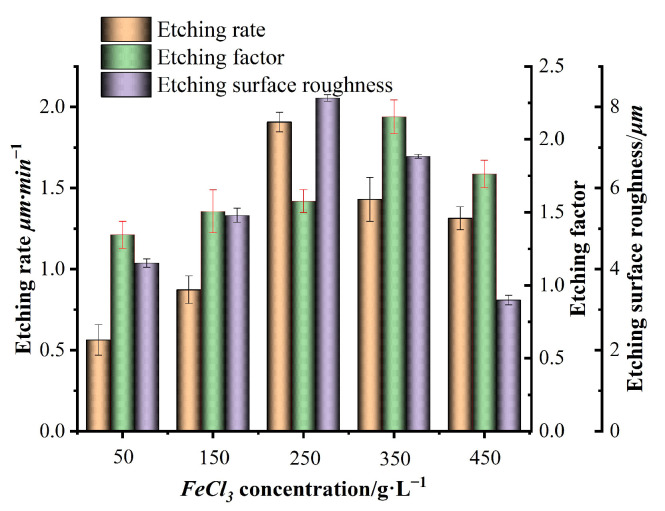
Effect of FeCl_3_ concentration on etching rate, etching factor, and roughness.

**Figure 4 micromachines-15-00929-f004:**
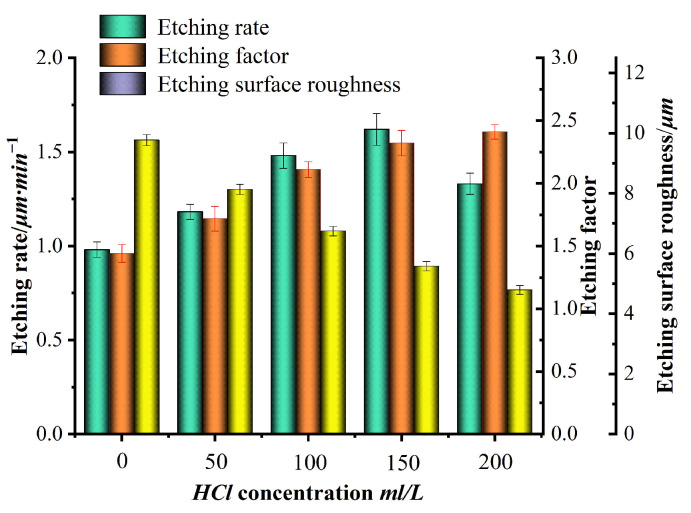
Effects of HCl concentration on etching rate, etching factor, and roughness.

**Figure 5 micromachines-15-00929-f005:**
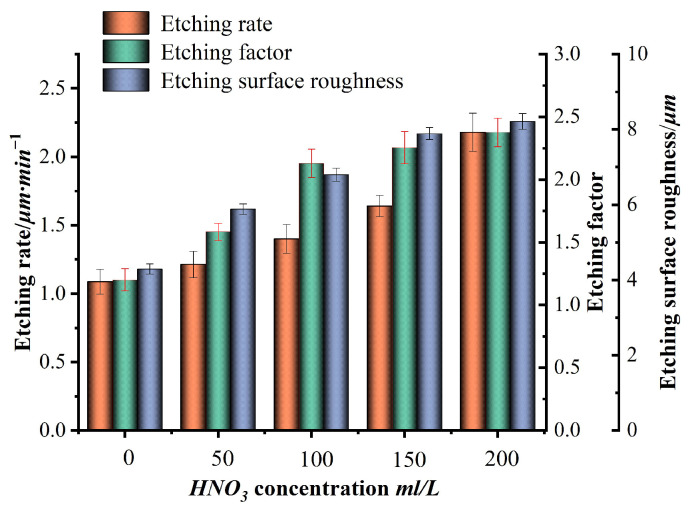
Effects of HNO_3_ concentration on etching rate, etching factor, and roughness.

**Figure 6 micromachines-15-00929-f006:**
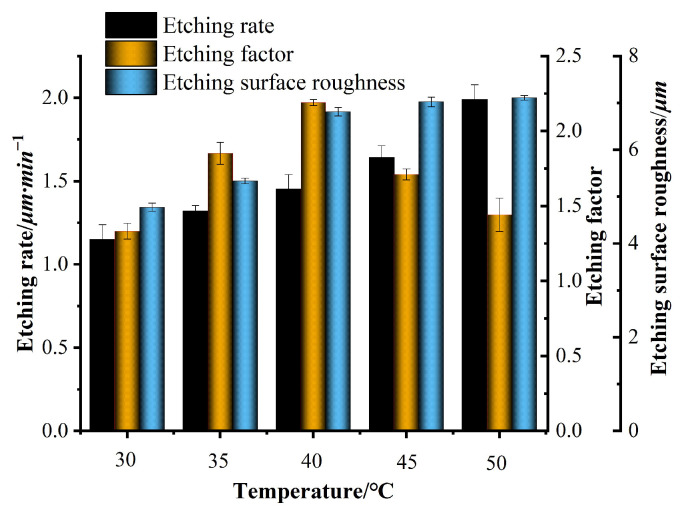
Effects of temperature on etching rate, etching factor, and roughness.

**Figure 7 micromachines-15-00929-f007:**
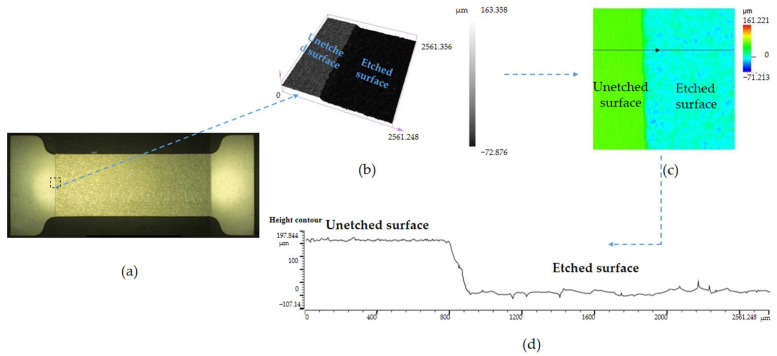
Surface morphology and etching depth map of 304 stainless steel substrate after wet etching, including the (**a**) Surface macroscopic diagram, (**b**) 3D scanning image of black square area, (**c**) Plan view of 3D scanning image, (**d**) Height difference map of black square area.

**Figure 8 micromachines-15-00929-f008:**
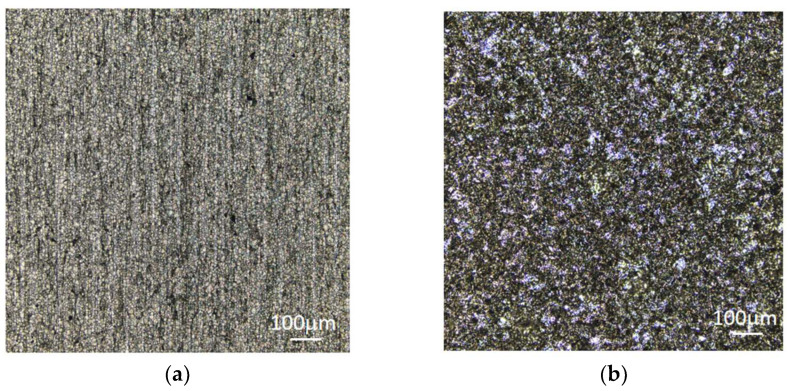
Optical microscope images of 304 stainless steel substrate surface before and after wet etching.

**Figure 9 micromachines-15-00929-f009:**
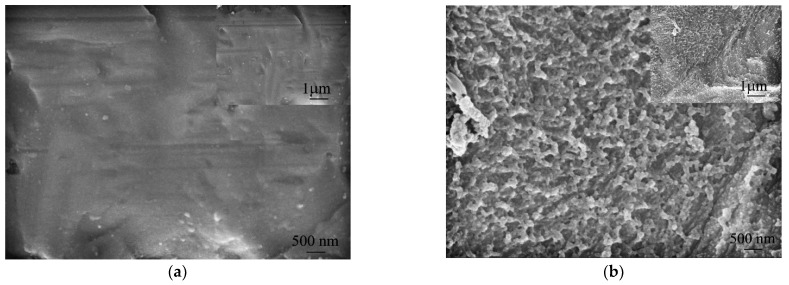
Scanning electron microscopy images of 304 stainless steel substrate surface before and after wet etching.

**Figure 10 micromachines-15-00929-f010:**
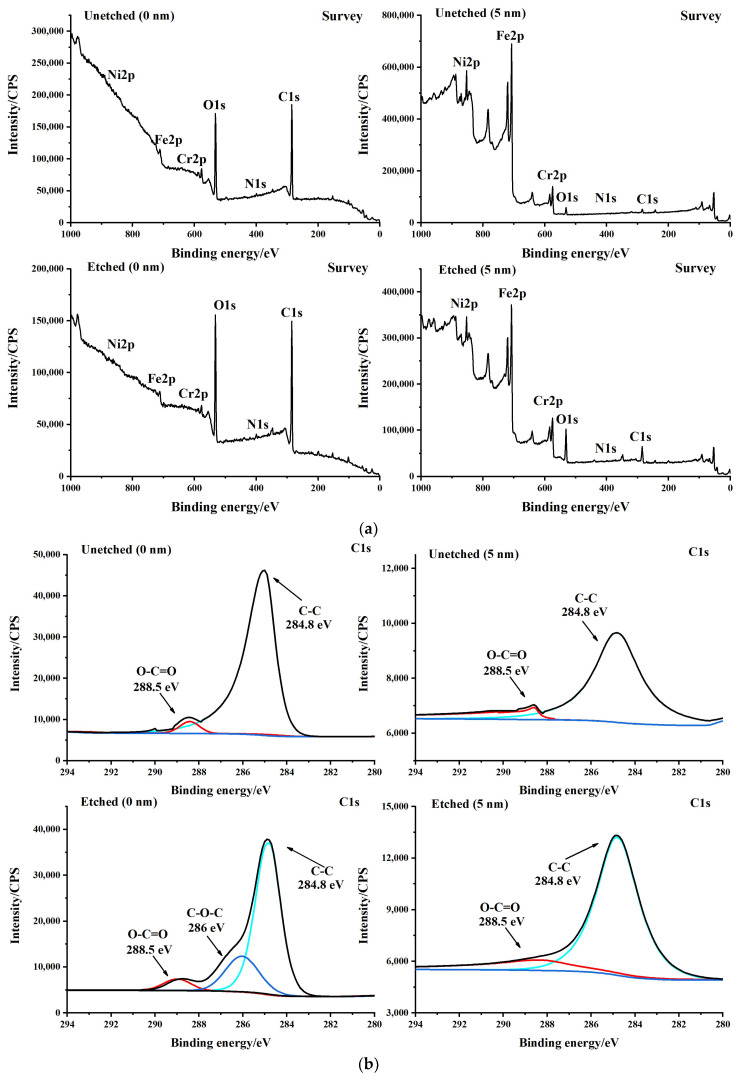
X-ray photoelectron spectra of the 304 stainless steel substrate surface at 0 nm and 5 nm before and after wet etching, including the (**a**) XPS total spectrum, (**b**) XPS spectrum of C1s, (**c**) XPS spectrum of O1s, (**d**) XPS spectrum of N1s, (**e**) XPS spectrum of Fe2p, (**f**) XPS spectrum of Ni2p, and (**g**) XPS spectrum of Cr2p.

**Figure 11 micromachines-15-00929-f011:**
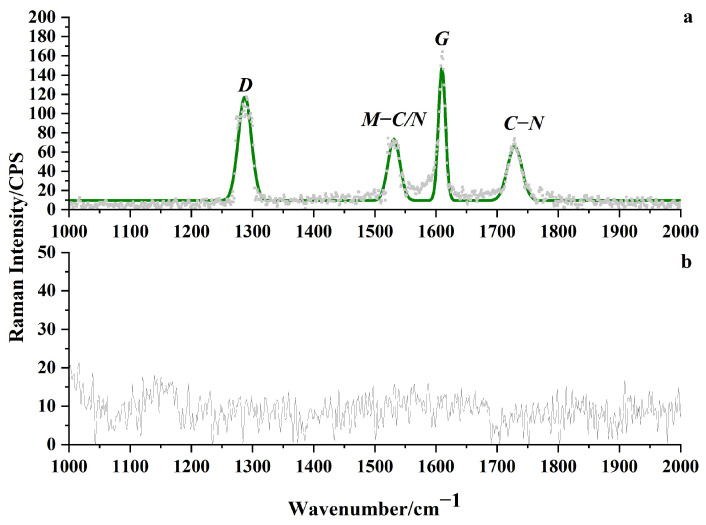
The Raman spectroscopic analysis of the surface of the 304 stainless steel substrate in the etched and unetched regions, where (**a**) is the etched region and (**b**) is the unexcavated region.

**Figure 12 micromachines-15-00929-f012:**
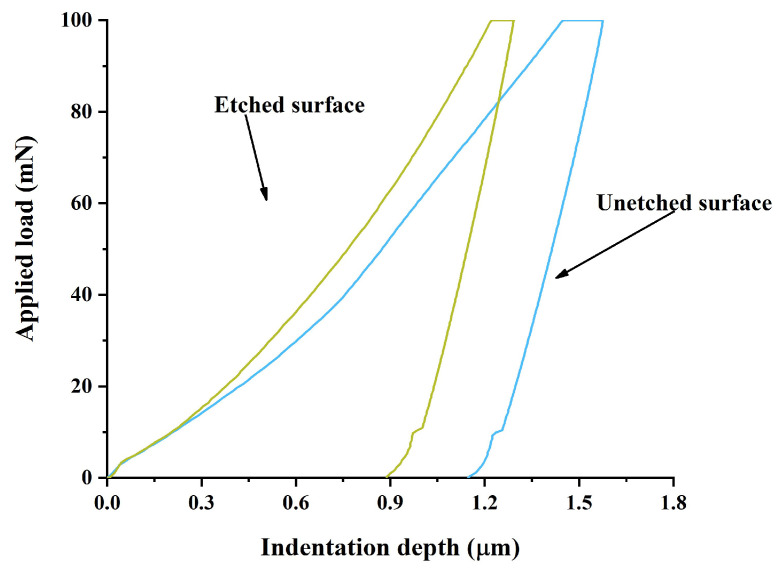
Indentation depth/applied load curve of nanoindentation under maximum load of 100 mN.

**Figure 13 micromachines-15-00929-f013:**
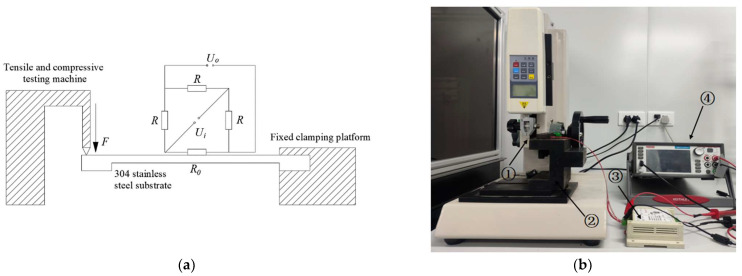
Bending test results of elastic substrates before and after wet etching treatment, including the (**a**) Test schematic diagram and (**b**) Test device diagram.

**Table 1 micromachines-15-00929-t001:** Test Samples and Etching Reagents.

Experimental Materials and Reagents	Molecular Formula	Specifications	Manufacturer	Location
304 stainless steel	SUS304	-	Shenzhen Deli Machinery and Hardware Co., Ltd.	Shenzhen, China
Ferric chloride	FeCl_3_	analytical pure	Chengdu Chron Chemical Co., Ltd.	Chengdu, China
Hydrochloric acid	HCl	analytical pure	Chengdu Chron Chemical Co., Ltd.	Chengdu, China
Nitric acid	HNO_3_	analytical pure	Chengdu Chron Chemical Co., Ltd.	Chengdu, China
Anhydrous ethanol	CH_3_CH_2_OH	analytical pure	Chengdu Chron Chemical Co., Ltd.	Chengdu, China
Photoresist	AZ6130	analytical pure	Changsha Jinxin Electronic Materials Co., Ltd.	Chnagsha, China

**Table 2 micromachines-15-00929-t002:** Test Analysis Instruments.

Instrument Name	Model	Company Name	Location
Precision electronic balance	JA603N	Shaoxing Pu’er Instrument Equipment Co., Ltd.	Shaoxing, China
Laser confocal microscope	OLS5000-SFA	Olympus Corporation	Tokyo, Japan
Optical contour analyzer	Contour GTK	Bruker Nano Surfaces Company	Goleta, CA, USA
Ultra-depth of field optical microscope	VHX-600	Keyence Corporation	Osaka, Japan
Scanning electron microscope	Tescan Mira4	Hitachi	Tokyo, Japan
X-ray photoelectron spectrometer	250Xi	Thermo Fisher Scientific	Norristown, MA, USA
Raman spectrometer	LabRAM Odyssey	Horiba	Kyoto, Japan
Nanoindentation instrument	NanoTest Vantage	Micro Materials Corporation	Wrexham, UK

**Table 3 micromachines-15-00929-t003:** Mechanical properties obtained from 100 mN nanoindentation test under maximum load.

304 Stainless Steel Substrate	Maximum Depth of Pressure (μm)	Maximum Load (mN)	Simplified Modulus (GPa)	Elastic Modulus (GPa)	Nano-Hardness (GPa)	*ζ* (%)
Unetched surface	1.574	100	182 ± 7	197 ± 9	2.17 ± 0.11	29.52
Etched surface	1.286	100	216 ± 11	242 ± 14	2.68 ± 0.13	27.46

**Table 4 micromachines-15-00929-t004:** Comparative analysis of theoretical values, simulation values, and bending test values.

Load (N)	Rectangular Substrate			I-Shaped Substrate			Ratio		
	Theoretical Value	Simulation Value	Experimental Value	Theoretical Value	Simulation Value	Experimental Value	Theoretical Ratio	Simulation Ratio	Experimental Ratio
0.2	1.86 × 10^−5^	1.73 × 10^−5^	(1.53 ± 0.02) × 10^−5^	7.36 × 10^−5^	6.9 × 10^−5^	(6.07 ± 0.74) × 10^−5^	3.96	3.99	3.97 ± 0.53
0.4	3.71 × 10^−5^	3.49 × 10^−5^	(2.99 ± 0.15) × 10^−5^	1.47 × 10^−4^	1.38 × 10^−4^	(1.01 ± 0.11) × 10^−4^	3.96	3.95	3.4 ± 0.54
0.6	5.57 × 10^−5^	5.23 × 10^−5^	(4.54 ± 0.19) × 10^−5^	2.21 × 10^−4^	2.07 × 10^−4^	(1.72 ± 0.13) × 10^−4^	3.97	3.96	3.8 ± 0.445
0.8	7.42 × 10^−5^	6.93 × 10^−5^	(6.67 ± 0.22) × 10^−5^	2.95 × 10^−4^	2.75 × 10^−4^	(2.51 ± 0.15) × 10^−4^	3.98	3.97	3.77 ± 0.35
1.0	9.28 × 10^−5^	8.67 × 10^−5^	(8.11 ± 0.17) × 10^−5^	3.68 × 10^−4^	3.45 × 10^−4^	(3.14 ± 0.12) × 10^−4^	3.97	3.98	3.88 ± 0.23

## Data Availability

The data that support the findings of this study are available from the corresponding author upon reasonable request.

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
