# Peer review of "Study on Electrical and Mechanical Properties of Double-End Supported Elastic Substrate Prepared by Wet Etching Process"

_micromachines, 2024, doi:10.3390/mi15070929_

Round 1

Reviewer 1 Report

Comments and Suggestions for Authors

1.        The advantages of wet eching process should be stated in detail in line 52.

2.        More references about stainless steel wet eching process may be added in the introduction section.

3.        In the introduction section, the second paragraph may be divided into two separate paragraphs.

4.        The tile of section 2 may be revised to “materials and methods”.

5.        In line 70-71, what is measured should be indicated specifically.

6.        “Before and after etching”, this expression is ambiguous, it should be stated in detail. How did the measure was conducted before and after etching? The experimental procedure was inadequate.

7.        What is quality method?

8.        The expression in Line 87 should be revised.

9.        In the abstract section the authors noted that laser cutting and wet chemical etching were combined. However, laser cutting has not been used in the paper. Lithography may be more accurate.

10.     Figure 10 should be revised for better understanding.

Comments on the Quality of English Language

Extensive editing of English language required.

Reviewer 2 Report

Comments and Suggestions for Authors

Song and Wu’s work, “Study on Electrical and Mechanical Properties of Double-end Supported Elastic Substrate Prepared by Wet Etching Process,” is an experimental study that describes the fabrication and electrical and mechanical properties of supported elastic substrates.
The work, well-written and organized, presents results that not only contribute to the field of micromachines but also hold the potential to inspire further research and development in this area.
I do recommend its publication after the authors review the following.

1) For non-specialist readers, could you please better explain the meaning of micro nanostructures?
2) When citing experimental equipment, you display the model of the equipment, company name, and location.
3) For non-specialist readers, please define the metal M in the stainless steel.
4) Figure 10 must be shorter and easier to understand while reading. I suggest breaking it down into parts and enhancing the caption figures, as it is too short to be meaningful.
5) Typo on page 18, line 467 (There is no Figure 3-16(a))
6) Figure 13 labels need to be appropriately placed.
7) In Tables 1 and 2, please add the experimental errors if possible.
8) Please add the ANSYS simulation conditions and the parameters used.
9) Can the author comment on the mechanical frequency properties of these structures?

Reviewer 3 Report

Comments and Suggestions for Authors

This work studies the impact of wet etching on electrical and mechanical properties of elastic substrate. By comparing the surface morphology, microstructure, and chemical and mechanical properties of 304 stainless steel substrate before and after etching treatment, the authors found that the height difference of the substrate surface before and after etching is between 160 μm and -70 μm, which is also confirmed by XPS and Raman spectroscopy. Overall, I find this work sound and clear, therefore recommend its acceptance after minor revision. Here are my comments.

1. Page 2, materials section, please present the chemical vendor information.

2. Page 6, figure 5 and 6, please rechoose color to make them readable for color-blind people.

3. In XPS and Raman data plot, the intensity was labeled as a.u., which is confusing. If they are arbitrary unit, then why are there numbers?
